# Investigating the Influence of Fluctuating Humidity and Temperature on Creep Deformation in High-Performance Concrete Beams: A Comparative Study between Natural and Laboratorial Environmental Tests

**DOI:** 10.3390/ma17050998

**Published:** 2024-02-22

**Authors:** Yixue Zhang, Jun Zhang, Jianghong Mao, Fei Lu, Zhuqing Jiang

**Affiliations:** 1Institute of Structural Engineering, Zhejiang University, Hangzhou 310058, China; yxzhang_zju@163.com; 2College of Civil Engineering and Architecture, Ningbo Tech University, Ningbo 315100, China; zj@nit.zju.edu.cn; 3College of Architecture & Environment, Sichuan University, Chengdu 610065, China; 4Huzhou Traffic & Plan Design Institute, Huzhou 313000, China; 18401606195@163.com; 5248 Geological Brigade of Shandong Nuclear Industry, Qingdao 266041, China; 11812046@zju.edu.cn

**Keywords:** HPC beams, creep, varying ambience, influence function, humidity, temperature, bridge

## Abstract

To investigate the influence of temperature and humidity variations on creep in high-performance concrete beams, beam tests were conducted in both natural and laboratory settings. The findings indicate that the variations in creep primarily stem from temperature changes, whereas humidity changes have little influence on fluctuations in both basic creep and total creep. The influence of humidity on creep is more strongly reflected in the magnitude of creep. Functions describing the influence of temperature and humidity on the creep behavior of high-performance concrete (HPC) subjected to fluctuating conditions are proposed. The findings were employed to examine creep deformation in engineering applications across four places. This study complements the correction method for the creep of members under fluctuating temperature and humidity. This research application can provide a basis for the calculation of the long-term deformation of HPC structures in natural environments.

## 1. Introduction

The serviceability of concrete constructions is contingent upon the delayed deformation resulting from creep. To ensure the reliability of civil engineering, a comprehensive understanding of creep behavior under sustained load and precise calculations are imperative [1,2,3,4]. Creep behavior holds particular significance for large-span and high-rise structures, wherein the utilization of high-performance concrete (HPC) is prevalent. However, existing creep prediction models for ordinary concrete (OC) exhibit substantial discrepancies when applied to the prediction of HPC creep [5,6,7]. Meanwhile, engineering structures often serve in a natural environment characterized by seasonally fluctuating temperature and humidity. According to Sakata and Ayano [8], environmental elements are crucial in the development of creep. Therefore, a specialized creep model for HPC that is applicable to fluctuating humidity and temperature conditions is required. The temperature/humidity level and temperature/humidity fluctuation are the primary indicators of temperature/humidity’s influence on creep. Most of the previous studies on creep were performed in environments with constant humidity and temperature but lack sufficient investigation into changing temperature and humidity settings. Furthermore, the majority of the research on the impact of temperature and humidity on creep focused on blocks and materials, while there is a dearth of research regarding components (e.g., beams, slabs and columns). Considering the potential influence of admixtures in the aforementioned studies is also essential for the study of HPC creep.

Investigations into the creep of OC impacted by ambient factors have been methodically carried out by using a significant volume of experimental data to establish a semi-theoretical and semi-empirical equation, along with various calculation specifications. Notably, the CEB FIP series [9,10,11,12], B3 [13], ACI 209 [14], GL-2000 [15], JTG 2018 [16], fib 2010 [17] and AASHTO [18] OC creep prediction models incorporate the consideration of ambient factors. It has been observed that creep escalates as humidity decreases [19,20] and temperature rises [20,21,22,23]. The mechanism is associated with basic creep and drying creep. Basic creep refers to the creep occurring when there is no moisture exchange with the environment, whereas drying creep is the total creep minus basic creep. It is generally believed that creep is attributed to the flow and deformation of calcium silicate hydrates (C-S-H), the flow and evaporation of pore water and the extension of microcracks [24,25]. Basic creep arises from the colloidal deformation of C-S-H and the flow of pore water, while drying creep is a result of the evaporation of pore water, leading to additional colloidal deformation of C-S-H and the flow of pore water. Unchangeable deformation results from the chemical reactions that accompany the aforementioned processes. Temperature-induced creep is caused by quicker drying [24,26] and faster deformation of C-S-H colloids [21,23]. The rise in the rate of water evaporation, along with the ensuing deformation and pore water flow, is the process responsible for the increase in creep with decreasing humidity [20,24]. It can be seen that the influence mechanism of temperature and humidity on creep is related to the mechanism of basic creep and drying creep; thus, temperature and humidity may affect basic creep and drying creep. The literature [27,28,29,30,31,32] discusses the impact of temperature on OC creep. The results from Hannant [27] and Nasser and Neville [30] revealed that rising temperatures clearly increased basic creep under high compressive strength, suggesting that HPC creep is more sensitive to temperature. Chen et al. [19] demonstrated that when the temperature cycle changed, the basic creep increased more noticeably in concrete with a high moisture content. Bazant et al. [26] found that concrete dries more quickly and has greater drying creep at higher ambient temperatures. Vidal et al. [23] proposed that thermal degradation might be the cause of the higher creep at high temperatures. According to Wei et al. [33], relative humidity is one of the main elements determining concrete creep, and concrete creeps less under sealed conditions. Manzoni et al. reported that nanoporous water dilation reduces C–S–H viscosity and consequently amplifies creep [21]. Chen et al. discovered that the amount of creep occurring at 100% ambient relative humidity was around one-third that in tests conducted at 60% ambient humidity [19].

Most creep studies so far have been under constant ambient conditions [20,34]. It is specified in [35] that the standard creep test should be carried out under standard laboratory conditions with a constant temperature (20 ± 2 °C) and constant humidity(60 ± 5%). Thus, the creep models established in the laboratory standard state can be taken as describing standard creep, with the others describing non-standard creep. Many studies [23,36,37,38,39,40,41,42,43] have indicated that alternating temperature and humidity affect creep; thus, bias occurs when applying laboratory results to engineering. Xu et al. [44] found that when the model parameters were derived from laboratory creep tests, the numerical simulations glaringly underestimated long-term settlement as compared to field measurements. Ladaoui et al. [22] and Vidal et al. [23] found that the transition from standard conditions (20 °C) to a higher temperature resulted in creep increasing by a ratio of 2.0 to 3.7. Tabatabai and Oesterle [38] found that the variation in creep and shrinkage owing to temperature changes only counteracts up to 15% of the temperature deformation; in the natural environment, the long-term deformation of concrete exhibits about the same oscillations as the variation in temperature. Schwesinger et al. [42] and Fahmi et al. [43] explored long-term deformation tests on axial compression and torsional members at varying temperatures. Illston and Sanders [45] considered the instantaneous temperature strain of creep at varying temperatures and obtained a creep prediction model suitable for alternating temperatures over 20 °C. Wang et al. [41] divided the ratio of the creep coefficients between natural and laboratory conditions into two parts—a stable part that was influenced by the average temperature and a variable part that was influenced by the temperature time course—and provided a model able to predict creep under simultaneous temperature and humidity variations. Many models are available to calculate the effects of creep at non-constant temperatures, such as BP-KX [26], B3 [13], B4 [46], CEB FIP90 [11], CEB FIP2010 [12] and Yang’s model [34]. However, the research mentioned above is still insufficient. All of them are based on specimens and materials, and there have been no examinations to determine whether the aforementioned creep rules apply to components. Differences in shape and force between members and test blocks make the concrete creep deformation in the components different from that in specimens, and the overall deformation of the member due to concrete creep is also affected by steel reinforcement. The above factors affect the application of the material creep law in the creep deformation of the component. In addition, many of the previous studies are based on OC rather than HPC.

Studies by Sakata and Ayano [8], Tabatabai and Oesterle [38] and Wang et al. [41] showed that varying temperature has a greater impact on creep fluctuation than does alternating humidity. Several investigations [34,47,48] have noted the particular creep behavior under varying humidity. Rahimi and Bazant [24] proposed that any change in relative humidity will enhance creep behavior. Hansen [36,37] argued that the creep in variable humidity is greater than that in a constant-humidity environment and only the first drying cycle could significantly increase the creep. After loading concrete specimens for a year under natural climatic circumstances, Vandewalle [48] found out that the total and basic creep were not clearly altered in response to changes in humidity. An analysis by Bazant and Wang [39] showed that alternating humidity mainly influenced the drying creep, and that the dependence of creep on temperature came from the hydration and creep activation energy. Based on the micro-prestressure consolidation theory [49], Bazant proposed a prediction model for creep under varying temperature and humidity. This model was further improved by Rahimi and Bazant [24], Gasch et al. [50], Wei et al. [25,33] and others. In addition, the Vidal model [23] and Yang model [34] describe creep modifications under alternating temperature and humidity. Yang et al. [34] also divided creep into a fundamental part and a fluctuating part. However, none of these have been examined at the component level and are likewise based on specimens. Quantification of the impact of fluctuating humidity on creep still needs to be improved.

Therefore, the HPC creep model must account for variations in humidity and temperature and be verified for use in HPC components. Currently, there are two approaches to HPC creep modeling. One method is modifying the OC creep model by applying an admixture influence function [4,5,6,7,51]. The other is directly constructing an HPC creep model through equal-proportion experiments [3,4,52]. Most of these two types of HPC creep models are based on constant surroundings, and further creep studies in environments with varying temperature and humidity are needed. The former HPC model directly adopts the regularity of the effect of temperature and humidity on the creep of OC. The latter HPC model arises from standard experimental conditions, so it cannot reflect the creep law under non-standard temperature and humidity conditions. Some HPC models from the latter group use the regularities for OC [4,6,7,19,51]. Their HPC creep formulae were built by establishing the HPC standard creep function and the influence function of ambient factors under non-standard conditions based on the CEB FIP2010, GL-2000, CEB FIP90, CEB FIP70 and fib 2010 models, respectively. However, aside from Wang et al. [4] comparing data from the HPC database, the majority of the influence functions were primarily calibrated using OC code, and their capability for calculating HPC creep remains questionable [22]. Admixture addition lowers the W/B ratio, allowing HPC to form more C–S–H and achieve better compactness [20,22]. This may change the pattern of OC creep with regard to temperature and humidity. For instance, Liang et al. [20] showed that concrete mixed with volcanic ash has more fine pores due to greater formation of C-S-H, which increases the water retention capacity compared to that of OC. Ladaoui et al. [53] found that HPC creep was more sensitive than OC creep under identical temperature rises. Ladaoui et al. [53] proposed that secondary C-S-H formation due to silica fume contributed to the porosity reduction and finer porosity of HPC, which accounted for the lower basic creep of the HPC. Given the disparities between OC and HPC, it is doubtful whether the regularities of ambient effects on OC creep can be extrapolated to HPC [22]. Thus, during execution, the application must be validated.

As shown above, the effect of temperature and humidity on concrete creep and its correction methods have been studied extensively, but corrections for the effect of varying temperature and humidity on creep remain inadequate. The rules of OC regarding the effect of temperature and humidity on creep may not apply to HPC due to the presence of admixtures, and it is yet unknown whether the temperature and humidity correction method applies to the members. Thus, research on the creep of HPC beams under constant and fluctuating temperature and humidity conditions was carried out in response to these doubts. The suggested correction technique will contribute to accurately modeling the creep of HPC in natural environments.

## 2. Basis in Theory

There have been studies on the effect of temperature and humidity levels and fluctuations on OC/HPC creep at the specimen and material levels, and specific influence functions have been proposed. Applying the above findings to temperature and humidity adjustments of component creep is also possible. This section first introduces the requirements that must be fulfilled for this application. Furthermore, the applicability of influence functions based on OC or temperature/humidity levels to the effect of fluctuating temperature and humidity on the HPC creep, and how to combine the influence factors of fluctuating temperature and humidity, is explored experimentally.

### 2.1. Form and Assumption of a Creep Function Considering Influencing Factors

One of the ways to establish a creep model that takes into account various influencing factors is choosing a benchmark condition to build a benchmark creep function, and then modifying it to obtain creep functions for non-benchmark conditions. For OC creep, this method has been incorporated into several models and standards. Codes ACI 209 [14] and CEB FIP70 [9] are typical examples of this methodology. The creep coefficients are expressed as the product of a series of coefficients, each of which represents an effect of a factor’s influence on creep, with every effect function independent of the other factors. This approach has also been employed in HPC creep modeling. Wang et al. [4], Su and Li [6], Guo et al. [7], Chen et al. [19] and Wang et al. [41] established HPC creep coefficient formulae under non-standard circumstances and attained a high degree of forecast accuracy. In accordance with the above studies, this methodology was also chosen in this paper to simulate creep considering temperature and humidity variations. The effect functions of varying temperature and humidity on creep are constructed separately. It is assumed that temperature and humidity impact creep independently of each other. Experiments were conducted to inspect this hypothesis.

### 2.2. Influence Function of Temperature and Humidity on Creep

The independent adjustment coefficient functions of temperature *K*(*T*) and humidity *K*(*H*), utilized for correction from benchmark creep coefficient $\varphi_{0}$ to creep coefficient φ, are provided based on the assumptions in Section 2.1. Hermite [19] established a linear equation of *K*(*H*) for the ratio of the creep coefficient between humidity *H* and benchmark humidity *H*_0_, as described in Equation (1), which decreases with *H*. $\varphi \left(H_{0}\right)$ is the creep coefficient in standard humidity; at other humidities, it is $\varphi \left(H\right)$, with the other settings kept the same as those in the standard creep laboratory.
(1)$$K\left(H\right)=\frac{\varphi \left(H\right)}{\varphi_{0}\left(H_{0}\right)}$$

In ACI 209 [14], the humidity-related influence coefficient of creep likewise falls linearly with *H*, but the model is only applicable when the humidity is higher than 40%. Su and Li [6] put forward a linear formula for the *K*(*H*) of HPC based on the GL 2000 [15] model, which decreases monotonically with *H*^2^. Wang et al. [4] set *H*_0_ at 95% and proposed the following:(2)$$K\left(H\right)=\frac{1}{0.70055-0.93949H+1.28201H^{2}}$$

Taking 100% relative humidity as *H*_0_, Equation (3) was proposed in CEB FIP70 [9].
(3)$$K\left(H\right)=1+3.25\left[1-e^{-0.01685\left(100-H\right)}\right]$$

The above functions were all put forward based on OC creep regulations. Therefore, it remains to be checked experimentally whether they are suitable for HPC. Additionally, the correction coefficients shown above are for constant humidity conditions. Whether they can be applied and how to apply them to an environment with changing humidity environment still needs to be investigated through experiments. A validation of their application to the components is needed as well. The *K*(*H*) functions given in the two codes have greater statistical confidence in terms of the sample size, and of these, Equation (3) has a larger application range in terms of humidity, so Equation (3) was chosen as the humidity influence function to test its application empirically.

A creep coefficient correction method is given in the CEB FIP90 [11] model for when the average temperature is outside the range of 10–20 °C. The creep coefficient function with temperature change is composited by correcting the nominal creep coefficient and the creep development coefficient, and by considering the instantaneous thermal creep coefficient at the moment of temperature increase, but this correction process is complicated and based on the law of OC. Yang et al. [34] suggested a temperature correction function that has been well validated by experimental results on HPC creep in the natural environment. Citing their findings, this paper argues that for every 1 °C increase, the creep coefficient increases by 0.004 times the squared temperature difference from the creep coefficient at *T*_0_ = 20 °C, and vice versa, taking a negative value, i.e.,
(4)$$K\left(T\right)=\varphi \left(T\right)-\varphi \left(T_{0}\right)$$
(5)$$K\left(T\right)=0.004\times {\left(T-20\right)}^{2}$$

*T* and *T*_0_ represent the temperature in nature and under standard conditions (20 °C), respectively. $\varphi \left(T_{0}\right)$ is the creep coefficient at standard temperature; at other temperatures, it is $\varphi \left(T\right)$, with the other settings kept the same as those in the standard creep laboratory. Equation (5) has already been used as an HPC creep adjustment function at fluctuating temperatures, so the only two verifications required are how to use it and its applicability to components.

According to Equations (3) and (5), the creep coefficient function for specimens, including the impact of varying temperature and humidity, can be set as Equation (6). The following fundamental assumptions are made when modeling.

(1)The strain is divided into a stable part, which is influenced by constant humidity and constant temperature, and a fluctuating part, which is influenced by variations in temperature.(2)Temperature decrease will lead to a reduction in the rate of creep as well as a reduction in the long-term deformation.(3)The influencing functions of each factor on creep are independent of each other.


(6)
$$\varphi \left(H,T\right)=K\left(H\right)\cdot \varphi_{0}+K\left(T\right)$$


$\varphi \left(H,T\right)$ is the creep coefficient in non-benchmark ambient conditions. In this study, experiments were conducted to investigate the applicability of Equation (6) to components and how to apply it.

### 2.3. Application of Specimen Creep Regulations to Components

The regularity of creep at the material level is subject to several assumptions and prerequisites before it can be applied to components. Taking a pure bending beam as an example, the analysis based on the theory of mechanics of materials [54] is as follows.

The beam can be identified as the Euler–Bernoulli beam pictured in Figure 1 if it is long, slender, and has a small height-to-span ratio.

The parameters are defined as follows: *x*, point *x* in the direction of the length of the beam; *y*, distance from the point on the cross-section to the neutral axis; $\theta_{0}\left(x\right)$, rotation of the cross-section at *x* caused by instantaneous elastic deformation; $\theta_{\varphi }\left(x,t\right)$, rotation of the cross-section at *x* caused by creep at time *t*; $\theta \left(x,t\right)$, total rotation of the cross-section at *x* and time *t*; *w*(*x*,*t*), the deflection curve equation of the beam when using the neutral axis of the beam as the x-axis; *s*, arc length; *ρ*, radius of curvature of a point on the neutral axis; *K*, curvature; ε, strain.

When the strain is small, it can be argued that $\theta =\tan\theta =\frac{dw}{dx}$. According to the definition, $\rho =\frac{ds}{d\theta }$; then,
(7)$$K=\frac{1}{\rho }=\frac{d\theta }{ds}\approx \frac{d\theta }{dx}=\frac{d^{2}w}{{dx}^{2}}$$

As $\bar{o^{’}o^{’}}$ is on the neutral axis, its length is $\bar{o^{’}o^{’}}=oo=bb$, so
(8)$$\varepsilon =\frac{\bar{b^{’}b^{’}}-bb}{bb}=\frac{\bar{b^{’}b^{’}}-\bar{o^{’}o^{’}}}{\bar{o^{’}o^{’}}}=\frac{\left(y+\rho \right)d\theta -\rho d\theta }{\rho d\theta }=\frac{y}{\rho }=yK$$

The following assumptions are made if the material creep coefficient is applied to the member directly:(I)The tensile creep is consistent with compression creep;(II)The plane section assumption is adhered to in the creep process;(III)The position of the neutral axis in the cross-section remains unchanged during creep.

As $\varepsilon (x,t)=\varepsilon_{0}(x)+\varepsilon_{\varphi }(x,t)$, we have
(9)$$\frac{1}{\rho (x,t)}=\frac{1}{\rho_{0}(x)}+\frac{1}{\rho_{\varphi }(x,t)}$$
(10)$$K(x,t)=K_{0}(x)+K_{\varphi }(x,t)$$
where the subscript “0” indicates the ε, *ρ*, and *K* values of the beam cross-section due to elastic deformation, and the subscript “φ” indicates the ε, *ρ*, and *K* values of the beam cross-section due to creep deformation. This notation also applies to Equations (11)–(14).

According to the definition, $\varepsilon_{\varphi }(x,t)=\varphi (t)\varepsilon_{0}(x)$. Thus,
(11)$$\frac{1}{\rho_{\varphi }(x,t)}=\varphi (t)\frac{1}{\rho_{0}(x)}$$
(12)$$K(x,t)=[1+\varphi (t)]K_{0}(x)$$

Then
(13)$$w(x,t)=\int \int_{0}^{x}K(x,t)dxdx=[1+\varphi (t)]w_{0}(x)$$
(14)$$w_{\varphi }(x,t)=\varphi (t)w_{0}(x)$$

Only then will the creep strain coefficient of the beam (written as $\varphi_{\varepsilon }$) and the creep deflection coefficient of the beam (written as $\varphi_{w}$) coincide. If $\varphi_{\varepsilon }$ is further equal to the creep in the specimen, then it is feasible that the creep law of the specimen applies to the calculation of the component creep deflection. At that time, the coefficient of influence of temperature and humidity on the creep coefficient in Section 2 will naturally apply to the component creep deflection coefficient. It is necessary to confirm whether these assumptions hold in practice.

It is also possible that the temperature and humidity adjustment functions in Equations (3) and (5) apply to the component, while the benchmark creep coefficient function $\varphi_{0}$ does not apply to the benchmark creep deflection factor $\varphi_{w0}$. This scenario is plausible since this work assumes the independence of the impact coefficients in Equations (3) and (5) and the benchmark creep coefficient $\varphi_{0}$. Because the effect of temperature and humidity on creep is the main focus of this study, the temperature and humidity influence function for the beam will be the major emphasis of this article, and we will not look at the applicability of the benchmark creep coefficient to the beam.

Thus, based on what remains to be confirmed in the calculation of the creep of HPC components under fluctuating temperature and humidity, as described in Section 2.1, Section 2.2 and Section 2.3, beam creep experiments under constant and changing temperature/humidity conditions were designed.

## 3. Materials and Methods

### 3.1. Materials

The beams were made of concrete with a compressive strength of 70 MPa. The mix proportion is illustrated in Table 1. The concrete slump was 200~220 mm. The cementitious materials consisted of ordinary Portland cement (P·O 52.5), ASTM Class I fly ash, and S95 slag, with performance indices detailed in Table 2. The medium sand used was natural river sand with a fineness modulus of 2.6. The water-reducing agent was a polycarboxy late superplasticizer. The elastic modulus of the concrete was 48.4 GPa, as measured in the experiment.

### 3.2. Specimen Design

As shown in Figure 2, the beam size was designed as 180 mm × 100 mm × 1240 mm, with 20 mm set aside for support placement at either end. The reinforcement bars were HRB400-grade bars manufactured by Hongtai Steel Co., Ltd. located in the city of Taizhou, Jiangsu, China with a yield strength design value of 360 MPa. As shown in Figure 3, two steel bars with a nominal diameter of 12 mm were buried longitudinally inside the beam members.

### 3.3. Creep Loading

As shown in Figure 2, a four-point bending test was carried out for the beam creep tests. A load of 1.5 kN was applied at each loading point, turning the span into a purely bending portion. The mid-span deflection was detected by means of a dial gauge, as presented in Figure 2.

The test followed the Chinese code The standard for long-term performance and durability test methods of ordinary concrete (GBT 50082) [35]. The testing beams’ information and test conditions are listed in Table 3. Symbol “B” refers to a beam, and the subscripts “c” and “v” indicate whether the temperature/humidity of the environment in which the beam was located was constant or variable, respectively. Beams B_CC-1_ and B_CC-2_ were shaped in a typical curing room and then moved to a standard creep room, as shown in Figure 4a. Conversely, beams B_VV_ and B_CV_ were placed in a natural environment, which was an experimental hall with a roof, shown in Figure 4b; the experimental hall had good ventilation to the outside world and, hence, varying temperature and humidity. The instruments shown in Figure 5 were arranged around the beams to record the variations in the temperature and humidity. Beam B_CV_ was wrapped with moisturizer to prevent drying and to keep it in a variable-temperature environment only. Creep tests on the beams were carried out at the same time.

## 4. Results

### 4.1. Elastic Deformation

The elastic deformation experimental data and a comparison with the theoretical value are shown in Figure 6. Since elastic deformation is not long-term and the loading and deformation were completed within 5 min, it can be inferred that the environmental conditions had little effect on the elastic modulus of the concrete in the beam. Therefore, the experimental values of elastic deflection of the beams under different temperature and humidity conditions will be theoretically similar. The experimental result in Figure 6 also shows that the deflections of four beams are closed, which can confirm the above assumptions. But there are also differences in elastic deformation between four beams shown in Figure 6; these may be an accidental error, or the elastic modulus of concrete in the beam may have been affected by the difference in temperature and humidity between the laboratory and the natural environment. Since the difference was within 5% and within the acceptable accidental error range, it is interpreted as a chance bias.

The load value in this experiment was designed to be less than the beam cracking load, so that the cross-section would not produce excessive concrete withdrawal from the work due to cracking and a better integrity could be maintained. If the modulus of elasticity is the same as the test results in Section 3.1, the beam deformation behavior can be considered to follow the deformation law of an Euler–Bernoulli beam, stated in Section 2.3; then, the theoretical value of mid-span deflection under load and self-weight can be calculated by using the following formula:(15)$${\bar{w}}_{0}=\frac{6.81FL^{3}}{384EI}+\frac{5qL^{4}}{384EI}=143(\mu m)$$
where *F* is the concentrated load of 3 kN; *L* is the beam clear length of 1.2 m; *q* is the beam self-weight uniform load of 0.45 kN/m; *E* is the elastic modulus of concrete; and *I* is the cross-section moment of inertia, *I* = 1.5 × 10^−5^ m^4^.

As shown in Figure 6, the elastic deflection test findings are in close agreement with the theoretical values. Since the theoretical value of elastic deflection is derived from the mechanics-of-materials approach, it can be surmised that the neutral axis of the beam passes through the center of the cross-section form and the beam deformation follows the plane sectional assumption.

### 4.2. Creep Behavior

The creep deflection is presented in Figure 7. The two B_CC_ beams had similar creep deflection. The temperature and humidity of the natural environment during the experiment, which ran from 12 July to 12 March of the next year, are shown in Figure 8 and Figure 9. The relative humidity in nature was mostly greater than the standard humidity (60%). Since the ambient humidity was approximately 85% during the creep testing, especially during the rainy season, the creep deflection of B_VV_ was generally lower than that of B_CC_. Moisture exchange did not take place between beam B_CV_ and the surrounding air, so it could be considered to be at 100% relative humidity. Thus, the deflection of B_CV_ was less than that of B_VV_. Deflection rose as the humidity decreased. The creep of B_CV_ was about two-thirds that of B_VV_, in line with Chen’s findings [19]. The creep deflection curves were smoothest for B_CC_ and most volatile for B_VV_, suggesting that fluctuations in temperature and humidity cause fluctuations in creep. In contrast, the deflection curve of B_CV_ was smoother than that of B_VV_ but more volatile than that of B_CC_. This is because B_CV_ was kept under constant humidity but fluctuating temperatures.

Studies [28,31,33,38] have highlighted the effect of temperature changes on basic creep. B_CV_ was sealed, so its creep was only basic creep [19] and its creep fluctuation came only from temperature variations. Since the fluctuations in creep for B_VV_ and B_CV_ were roughly the same as shown in Figure 7, it can be inferred that in the natural environment, fluctuations in the basic creep due to temperature variations were the primary cause of fluctuations in the HPC creep under changing temperature and humidity, while the effect of alternating humidity on the basic creep was not evident. This experimental result in terms of beam deformation is consistent with the regularity of the specimen, which was discussed in research by Sakata and Ayano [8], Tabatabai and Oesterle [38], Wang et al. [41] and Vandewalle [48]. Sakata and Ayano [8] and Bazant and Wang [39] suggested that the contribution of humidity fluctuations to concrete creep can be ignored. Tabatabai and Oesterle [38] pointed out that the long-term deformation of concrete in a natural environment shows approximately the same fluctuations as it does with temperature. Vandewalle [48] showed that total and basic creep do not appear to vary in response to changes in humidity. Thus, it can be seen that the variations in creep mainly stem from temperature changes, whereas humidity changes have little influence on the fluctuation of both basic creep and total creep. The influence of humidity on creep is more strongly reflected in the magnitude of creep, as indicated by the phenomenon in Figure 7, which shows that the creep values of B_CC_, B_VV_ and B_CV_ were in the same order as the humidity. Additionally, the law of the effect of temperature and humidity fluctuations on the long-term deformation of the component is consistent with the law of the relevant research on materials reviewed in the introduction, which indicates that the research on the impact of varying temperature and humidity on long-term deformation under constant load at the material level is qualitatively applicable to components.

### 4.3. Creep Model for Beam Considering the Influence of Changing Temperature and Humidity

Equations (3), (5) and (6) in Section 2.2 describe the creep of the materials used in this specimen study. Their application to beams is verified in this section. Figure 10 displays the average creep coefficients of deflection for the two B_CC_ beams, denoted as $\varphi_{0}$. Using $\varphi_{0}$ modified by Equation (6), the theoretical values of φ for B_CV_ and B_VV_ were obtained. The applicability of Equations (3), (5) and (6) to components and the way of applying them were explored by comparing the theoretical values with the experimental values for B_CV_ and B_VV_.

When calculating *K*(*T*) and *K*(*H*) using Equations (3) and (5), the values of the variables *H* and *T* need to be determined. *H* and *T* are taken as the average temperature and humidity of the interval, affected by interval length. We define *i*_T_ and *i*_H_ as the time intervals used to compute the average values of the variables *H* and *T* for the calculation of *K*(*T*) and *K*(*H*) by means of Equations (3) and (5). For example, if *i*_T_ = 1 day, then we calculate the daily average temperature *T* and enter it into Equation (5) to obtain *K*(*T*). Thus, the values of *H* and *T* affect the values of *K*(*T*) and *K*(*H*) and, subsequently, the accuracy of the theoretical value calculated by means of Equation (6). It can be inferred that the larger *i*_T_ and *i*_H_ are, the smaller the fluctuations in *K*(*T*) and *K*(*H*) are over time. The calculated theoretical value of beam deformation will subsequently fluctuate less. For an interval duration of one day, the daily humidity correction factor *K*(*H*) calculated from Equation (3) is shown in Figure 8, and the daily temperature correction factor *K*(*T*) calculated from Equation (5) is shown in Figure 9.

The experimental results for beams and a review of research on specimens showed that the creep fluctuation is more compatible with the temperature fluctuation; therefore, the daily average temperature might be a suitable choice for T to calculate *K*(*T*) by means of Equation (5). For beam B_CV_, the fluctuation in modified creep values when using 100% humidity and the daily average temperature compares well with the experimental values, confirming the suitability of Equations (3) and (5) for the creep of HPC beams, along with the reasonableness of taking the daily average temperature to calculate *K*(*T*) when calculating the long-term deformation of beams at varying temperatures.

Humidity fluctuation has a smaller effect on creep fluctuation, and the humidity level has a larger effect on the creep magnitude. Therefore, *i*_H_ should not be too short; otherwise, it is likely to cause excessive fluctuations in the theoretical deflection of the beam. For *i_H_* values of 1 day and 40 days, the theoretical values for B_VV_ obtained by adjusting the experimental results for B_CC_ using Equations (3), (5) and (6) are shown in Figure 10. Compared with the experimental deflection for B_VV_, the adjustment results appear to be too volatile when using the daily average humidity to obtain *K*(*H*). Since the rate of change in the humidity in concrete is lower and slower than that in the surrounding environment, it was thought that extending *i_H_* would lessen the impact of fluctuations in humidity on the volatility of the creep and would comply with the mechanism of the effect of humidity on creep as demonstrated by the experimental results. Therefore, 40 days was set as the humidity interval *i*_H_, and *K*(*H*) was computed using the average humidity for each interval; the result shows better fitness, as can be seen in Figure 10.

## 5. Discussion

### 5.1. Mechanism Analysis

Humidity directly impacts the rate of evaporation of water; thus, it has a direct influence on drying creep and an indirect influence on basic creep. Temperature directly affects the rate of generation of C-S-H colloids and their deformation properties (e.g., Bazant and Wang [39] showed that temperature mainly affects the hydration and activation energy), thus having a more direct impact on the basic creep. As the C-S-H colloid has strength, changes in its instantaneous stiffness have a more notable impact on the short-term deformation properties, which is also reflected in the contribution of temperature changes to fluctuations in long-term deformation. The evaporation of pore water, on the other hand, involves material loss from the test specimen itself and plays a decisive role in the ultimate volume of the object. Water loss reduces the interlayer water between C-S-H molecules, increasing the chemical binding between C-S-H molecules and thus increasing irreversible creep deformation. This affects the quantities of dry creep and total creep. However, it is evident that the change in moisture evaporation both inside concrete and between concrete and the environment is a delayed process, so it does not have a relatively large effect on short-term deformation. Furthermore, the impact of a change in the humidity of the surrounding environment on the humidity inside concrete is lesser than that of outside temperature change on the temperature variation in concrete. Thus, the effect of changes in moisture on creep fluctuations can be ignored. Short-term fluctuations in long-term deformation originate from temperature changes. A study by Wang et al. [41] also argued for the above mechanism.

### 5.2. Impact of Different i_H_ and i_T_ Values on the Adjustment Results of Equation (6)

To investigate the optimal choices of *i*_T_ and *i*_H_, the experimental values for B_CV_ and B_VV_ were compared with their theoretical values calculated using different *i*_T_ and *i*_H_ values, as shown in Figure 11 and Figure 12. Since the experimental values of B_CV_ and B_VV_ deflection exhibited comparable fluctuations, and since B_CV_ was subject to constant humidity and variable temperature, the accuracy of *K*(*T*) can be verified by comparison with the experimental value for B_CV_. As shown in Figure 11, the theoretical results of B_CV_ beam deflection were obtained using Equation (5) with the fluctuating temperature *T* calculated for *i*_T_ = 1, 10 and 30 days under 100% humidity. The result shows that an increase in *i*_T_ weakens the impact of temperature fluctuation on fluctuations in the theoretical creep deflection, and it confirms that the choice of 1 day for *i*_T_ is reasonable when obtaining *K*(*T*).

On this basis, *i*_H_ was checked by contrast with the experimental values for B_VV_. As shown in Figure 12, the theoretical value for B_VV_ was calculated for *i*_T_ = 1 day and *i*_H_ = 1, 10, 30, 40 and 50 days. It was found that when the humidity interval was 1, 10 or 30 days, the creep fluctuation was too large, and when it was 50 days, the theoretical creep deflection showed less fluctuation than did the experimental value. It was further confirmed that choosing an *i*_H_ of 40 days is a fair decision when computing the influence coefficient *K*(*H*).

## 6. Engineering Application

The temperature and humidity adjustment method presented herein was used to calculate the 30-year creep deformation of the North Navigation Channel Bridge at Zhoushan Port shown in Figure 13; this was accomplished by modeling the bridge in the finite element software product Midas Civil 2019 (v2.1). The creep coefficients for both the naturally varying temperature and humidity environment and the designed common constant-temperature and -humidity environment were used for comparison.

The engineering structure is located in Ningbo, Zhoushan Islands, China. The HPC composition of the bridge is shown in Table 1. The bridge is a continuous girder bridge, designed as a two-way four-lane highway standard, with the piers firmly attached to the main girders. The main span is 260 m, with steel box girders in the middle of the span within 85 m. Each end is connected to the concrete section of the main girder by a 5 m steel-mixed section. The remaining 110 sections are precast segmental girders, which were constructed on-site by means of balanced cantilever lifting. The precast sections are mainly 3, 3.5, 4 and 4.5 m long and 4.8 to 12.6 m high, with the section height and bottom slab thickness varying parabolically by 1.6 times in the direction of the bridge. The main bridge elevation and box girder cross-section are shown in Figure 13.

As shown in Figure 14, a beam unit was used for each precast box girder section in the finite element model. The prestressing steel bundles are positioned symmetrically on both sides of the section. Fixed bearings are provided at both ends of the bridge. The piers are restrained to the girder sections above them by stiff arms. The steel box girders are connected elastically to the precast girder sections.

The custom creep function in the software was used to assign the time-dependent material properties calculated by means of Equation (6). Figure 15 displays the daily temperature and humidity readings that were recorded on-site at Zhoushan during a period of four years. These were extended as a set of cyclic units to 30-year temperature and humidity values. The creep coefficient was calculated using Equation (6) and entered into the bridge model; then, the 30-year deformation of the bridge under the environmental conditions in Zhoushan was calculated, as shown in Figure 16. In addition to Zhoushan, according to the statistical data from the China Meteorological Administration, the deformation of bridges in Beijing, Shanghai and Guangzhou was calculated and plotted together with the deformation of bridges calculated using the CEB FIP90 model under the designed constant temperature and humidity conditions (20 °C, 60%) in Figure 16. The maximum mid-span deflection under the naturally varying temperature and humidity in Zhoushan was 112 mm. The long-term deformation under the designed conditions was more conservative than that from the new model (Equation (6)).

The long-term deformation of bridges is greater in Beijing than in Zhoushan due to the lower humidity, whereas Guangzhou has higher humidity than Zhoushan and, thus, shows reduced long-term deformation of its bridges. The bridge deformation derived from the creep model established using the actual temperature and humidity conditions for the four locations is smaller than that produced by the CEB FIP90 model under the designed constant temperature and humidity settings.

## 7. Conclusions

Creep tests on beams under constant and variable temperature and humidity conditions were carried out. The main conclusions are as follows.

(1)Oscillations in creep are mainly caused by changes in temperature, whereas changes in humidity have little influence on fluctuations in both basic creep and total creep. The influence of humidity on creep is more strongly reflected in the magnitude of the creep.(2)A humidity adjustment factor and a temperature adjustment factor for creep were developed for application to high-performance concrete. These factors were validated at the component level and can accurately describe how temperature and humidity affect the amount of and fluctuations in creep.(3)The 30-year creep deflection of an engineering structure in a natural environment in four places was calculated by means of modeling in Midas Civil.(4)The mix ratio of the concrete material in this study was finite, and future validation should be carried out for a wider range of material types and component forms. The laws obtained so far are based on macroscopic empirical summaries. More theoretical and modeling studies at the microscopic scale are needed to reveal their essence.

## Figures and Tables

**Figure 1 materials-17-00998-f001:**
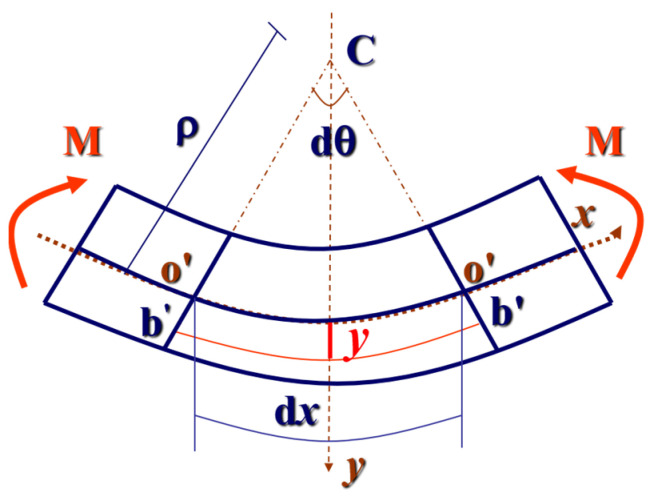
Bending of Euler–Bernoulli beam.

**Figure 2 materials-17-00998-f002:**
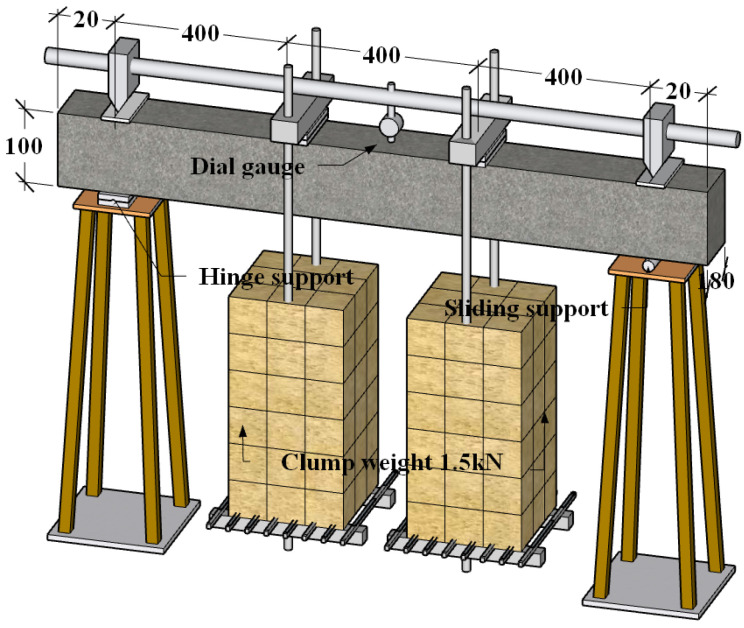
Schematic test setup and creep measurement (in mm).

**Figure 3 materials-17-00998-f003:**
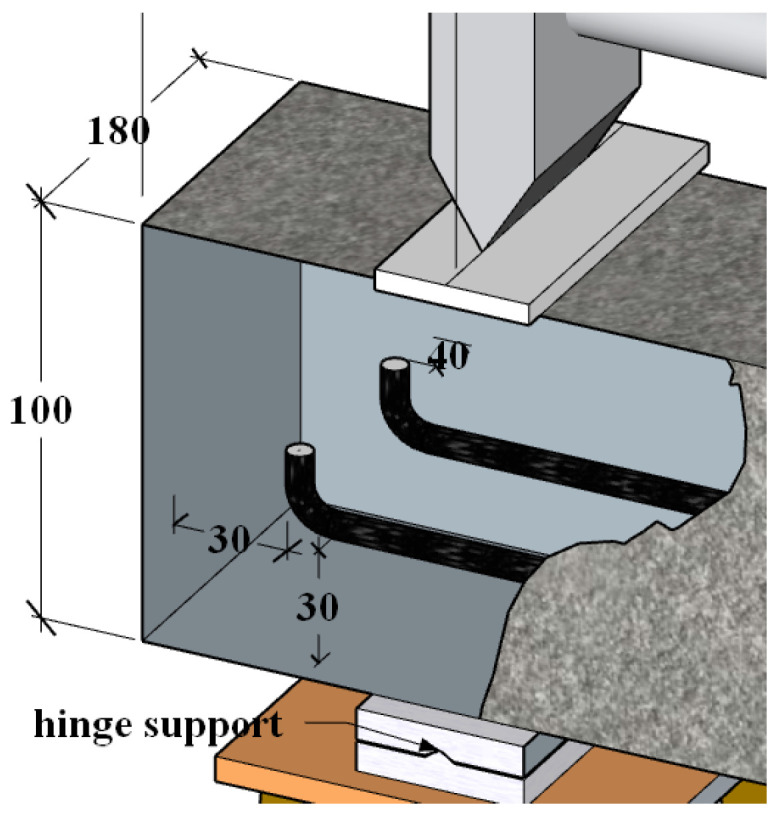
Details of the reinforced concrete (RC) beam specimen (in mm).

**Figure 4 materials-17-00998-f004:**
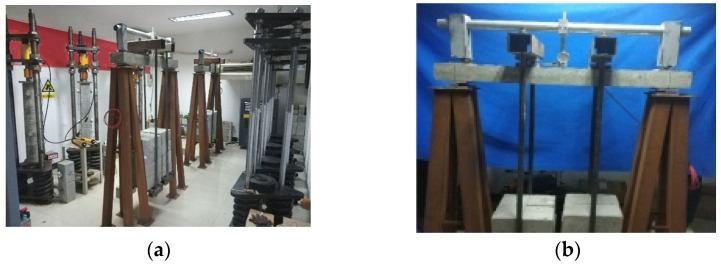
Ambient setup for the RC beam specimens. (**a**) Beam in constant ambient conditions. (**b**) Beam in natural conditions.

**Figure 5 materials-17-00998-f005:**
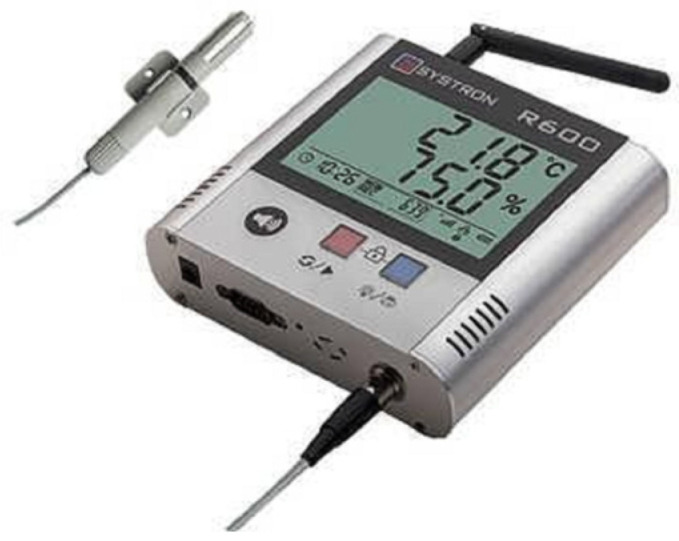
Temperature and humidity sensor.

**Figure 6 materials-17-00998-f006:**
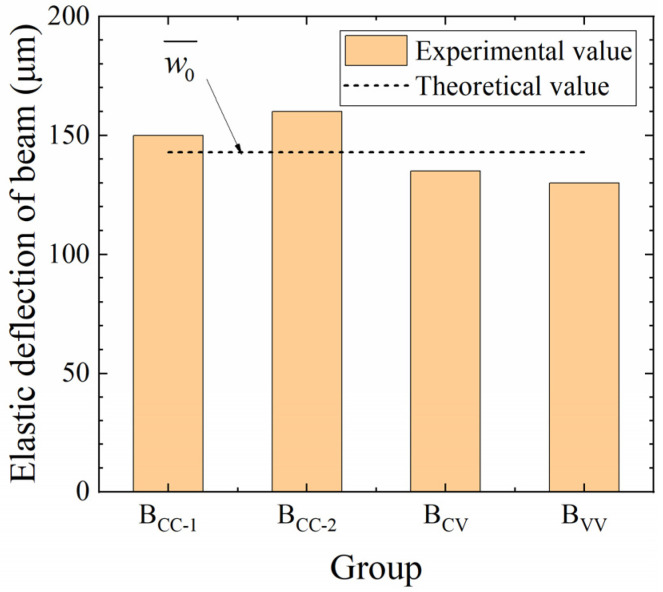
Experimental results and theoretical analysis of mid-span elastic deformation.

**Figure 7 materials-17-00998-f007:**
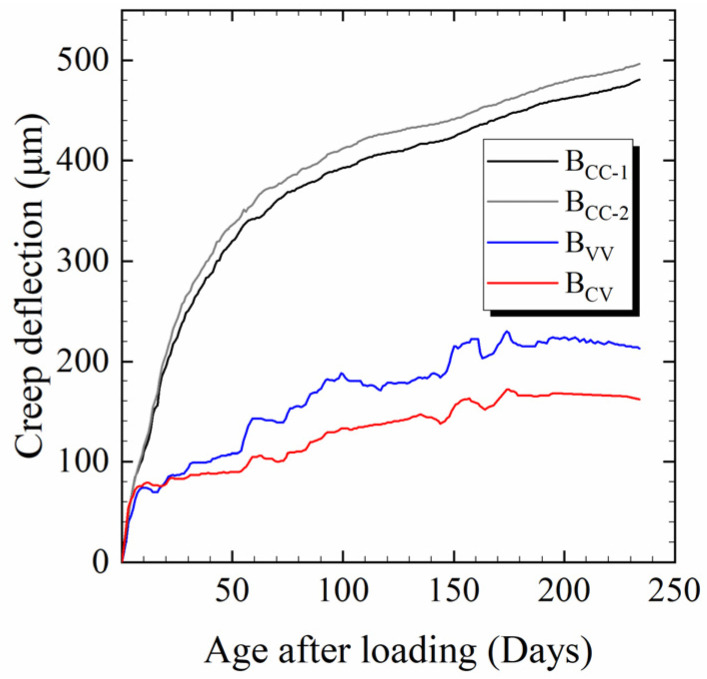
Creep deflection of the beam under the three sets of conditions shown in Table 3.

**Figure 8 materials-17-00998-f008:**
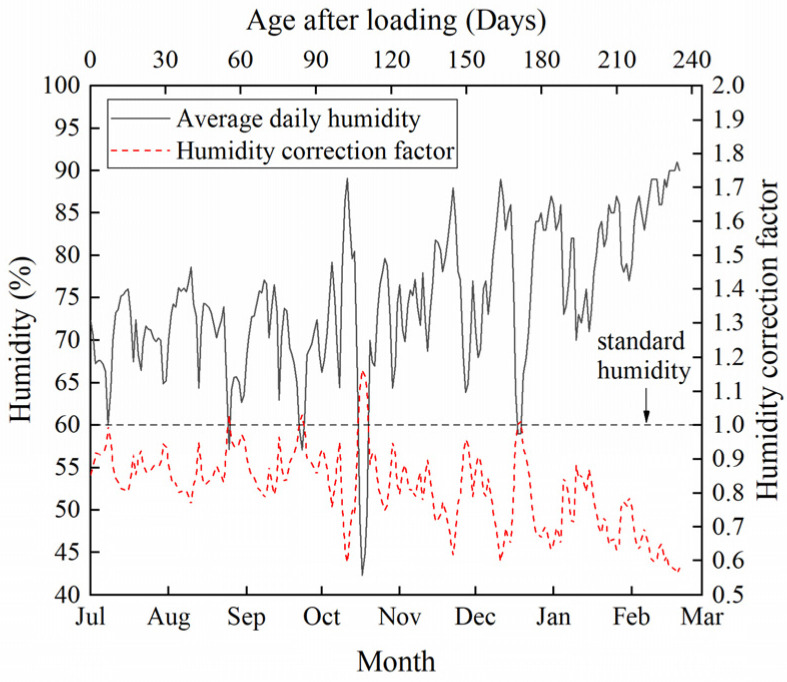
Humidity in nature and a corresponding correction factor for the creep coefficient.

**Figure 9 materials-17-00998-f009:**
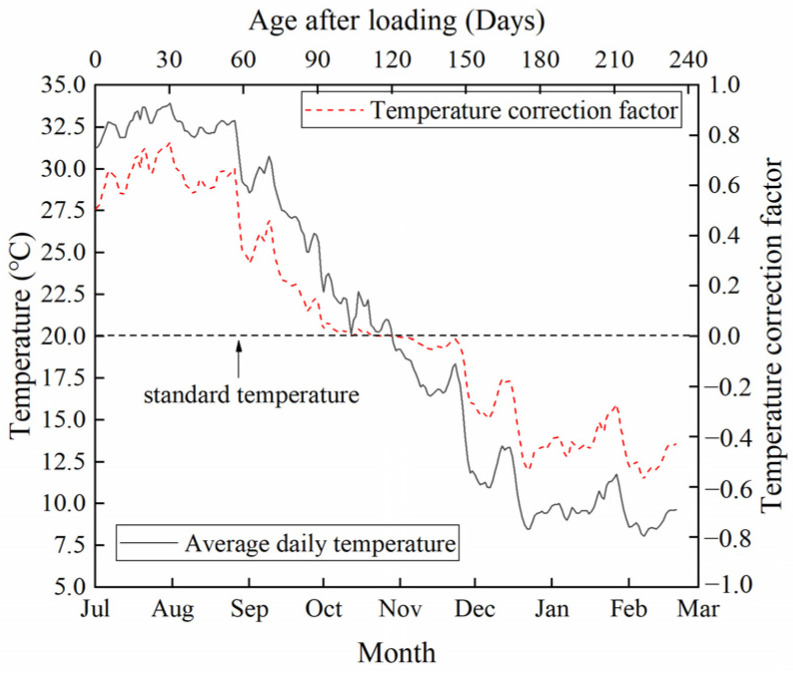
Temperature in nature and a corresponding correction factor for the creep coefficient.

**Figure 10 materials-17-00998-f010:**
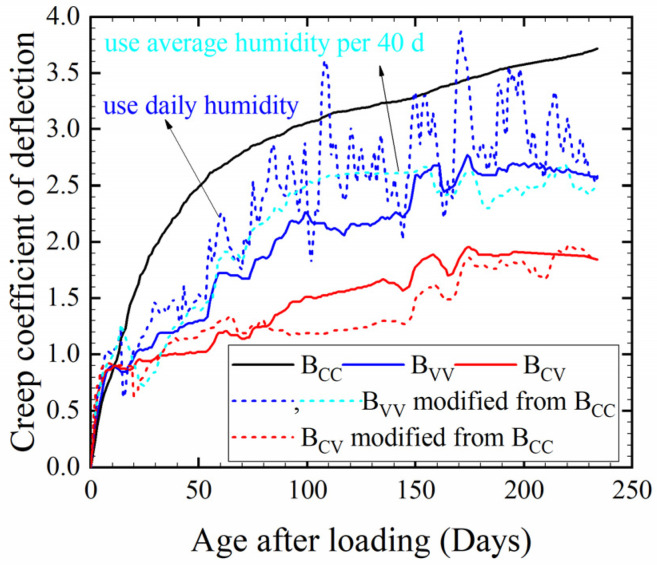
Environmental correction of the creep coefficient of deflection and experimental validation.

**Figure 11 materials-17-00998-f011:**
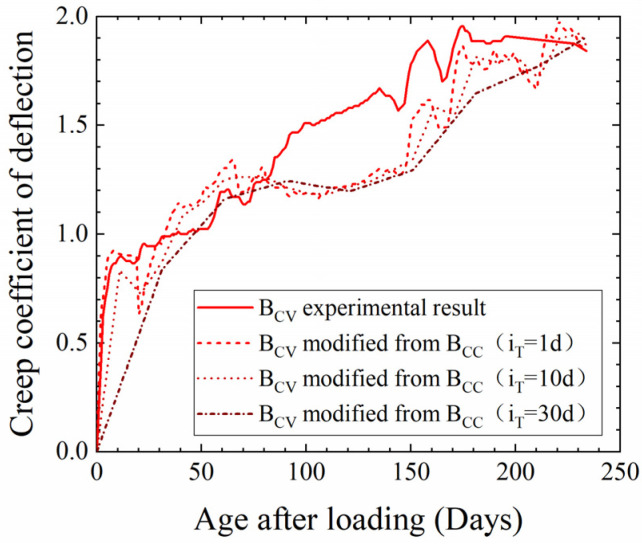
Theoretical results of B_CV_ creep deflection calculated using different *i*_T_ values and comparison with experimental results.

**Figure 12 materials-17-00998-f012:**
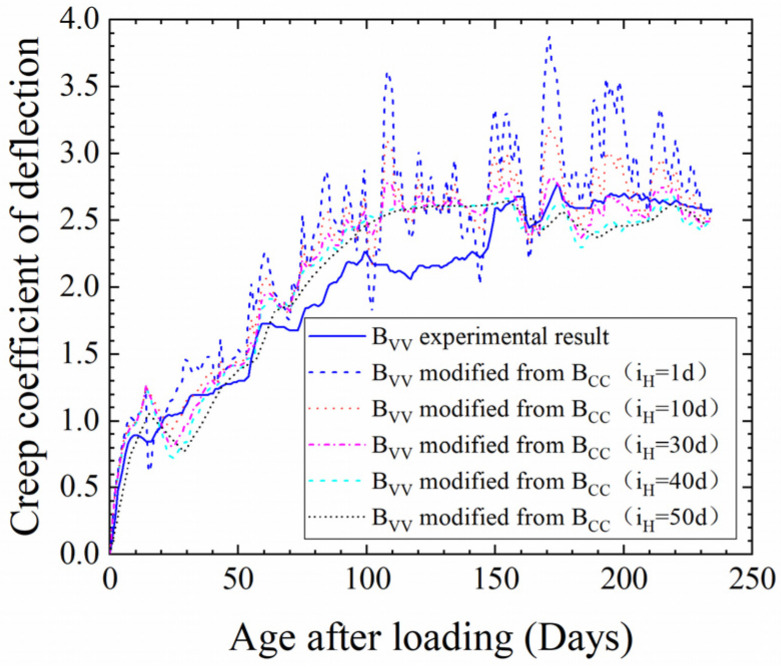
Theoretical results of B_VV_ creep deflection calculated using different *i*_H_ values and comparison with experimental results.

**Figure 13 materials-17-00998-f013:**
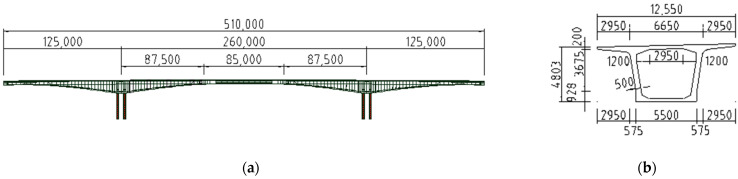
North Navigation Channel Bridge. (**a**) Elevation view of the North Navigation Channel Bridge. (**b**) Cross-section of a box girder.

**Figure 14 materials-17-00998-f014:**
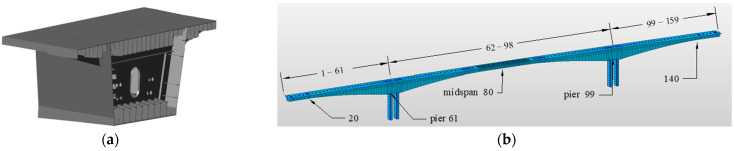
Bridge model and node diagram in Midas Civil. (**a**) Model cross-section. (**b**) Model nodes.

**Figure 15 materials-17-00998-f015:**
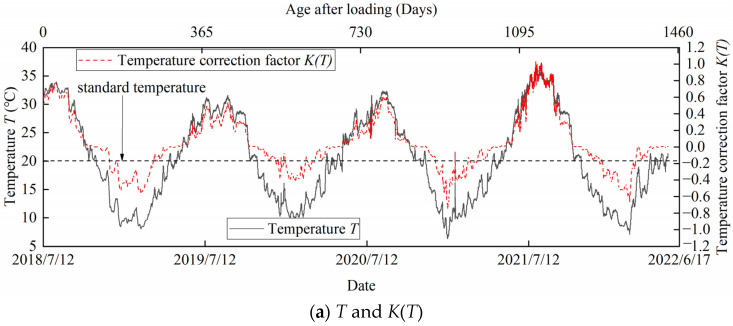
Daily temperature and humidity values collected on-site in Zhoushan over 4 years and the corresponding correction factors.

**Figure 16 materials-17-00998-f016:**
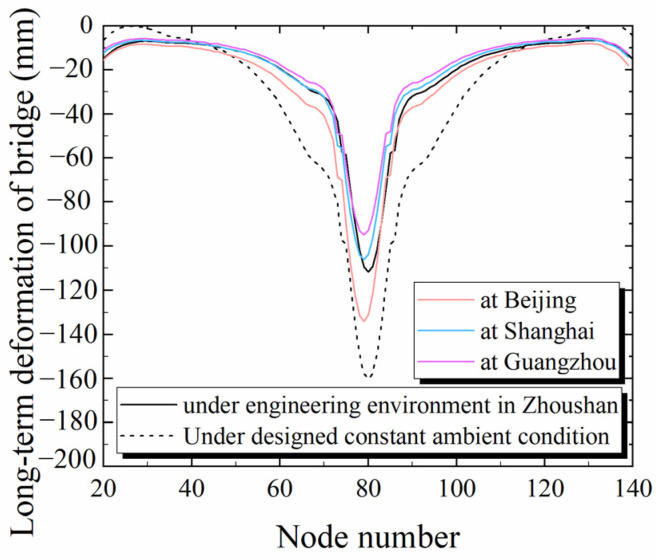
Long-term deformation of the North Navigation Channel Bridge.

**Table 1 materials-17-00998-t001:** Concrete composition.

Material Name	Cement	Sand	Crushed Rock		Slag	Fly Ash	Water	Water-Reducing Agent
			15–25 mm	5–15 mm				
Amount ($kg/m^{3}$)	490	700	419.6	629.4	69	69	156	4.9

**Table 2 materials-17-00998-t002:** Physical properties and chemical composition of slag and fly ash [55].

Admixture	Ignition Loss/%	Moisture Capacity/%	CaO/%	MgO/%	28 d Activity Index/%
Slag	0.53	0.6	-	5.86	97
Fly ash	3.36	0.4	0.28	-	98

**Table 3 materials-17-00998-t003:** Testing beam information and test condition.

Group	Beam Number	Test Conditions	Remark
B_CC_	2	Constant temperature and humidity	Beams in a standard creep test laboratory
B_CV_	1	Varying temperature and constant humidity	No-drying beam in a natural environment
B_VV_	1	Varying temperature and humidity	Beam in a natural environment

## Data Availability

Data are contained within the article.

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
