# Peer review of "Investigating the Influence of Fluctuating Humidity and Temperature on Creep Deformation in High-Performance Concrete Beams: A Comparative Study between Natural and Laboratorial Environmental Tests"

_materials, 2024, doi:10.3390/ma17050998_

Round 1

Reviewer 1 Report

Comments and Suggestions for Authors

This work investigates the influence of humidity and temperature on creep deformation in concrete beam.

The content of this manuscript is sufficient, and the findings provide certain information on explaining the difference between laboratory test and actual application.

More comments are given below:

1. The abstract is not detailed enough, since it does not highlight the significance of this work, and the reported findings are too generic.

2. Lines 102–103: “All of them are based on specimen and material, and there is a lack of examination to see if the aforementioned creep rules apply to components.” In this part, please explicitly elaborate the underlying causes of this concern. How do the creep rules differ specimen/material and structural components?

3. Section 3 (Experiments): Please provide a table showing test programme.

4. Section 4.1 (Elastic deformation): The authors should explain the difference in the results between BCC-1, BCC-2, BCV, and BVV. Also, why four specimens are compared to only a single theoretical value? Do these four specimens undergo similar testing conditions, such that its elastic deformation must all be in close agreement with a single theoretical value?

5. The conclusion should be rewritten to be more concise and present only the main findings.

Comments on the Quality of English Language

The English language should be polished.

Author Response

Thank you very much for taking the time to review this manuscript. Please find the detailed responses below and the corresponding revisions highlighted changes in the re-submitted files.

Reviewer 2 Report

Comments and Suggestions for Authors

General Comment

The submitted manuscript presents a study to investigate the influence of both temperature and humidity fluctuations on the creep behavior of high-performance concrete (HPC) beams. After a literature review on the topic, where the authors justify the need for more studies on the creep deformation models for HPC members, a review on models based on influence functions is presented and a model is proposed for HPC beams in order to include both temperature and humidity fluctuations. To check the validity of the model, an experimental program involving 4 test HPC beams was performed. Two beams were tested in the laboratory condition (with temperature and humidity controlled), and the other two were tested in natural environment (with temperature and temperature + humidity fluctuation). The materials, test specimens and testing procedures are described. The results are presented and discussed, in terms of the influence of both temperature and humidity fluctuations on the creep behavior of the test beams, and also to assess the predictions of the proposed model. The authors conclude about the accuracy of the model. Finally, to demonstrate its applicability, the model is applied to an existing bridge across four cities in china. The results are presented and discussed.

The topic of the manuscript is interesting and actual, since models to predict the creep deformation of HPC members still need to be studied. The consideration of both fluctuating humidity and temperature constitute an additional value for this study. The proposed model and results from this study could be useful for future researches and also for future standards revisions.

I made some comments/suggestions in order to improve the manuscript. The authors should take the comments into account and revise their manuscript.

Specific Comment 1

The article must be entirely revised by a professional to improve the reading and correct typos.

Specific Comment 2

Title

Consider writing “HPC Beams” instead of “Components”, and also “Natural and Laboratorial Environmental Tests”.

Specific Comment 3

Keywords

Add “HPC beams” as the first keyword.

Specific Comment 4

The citations in the manuscript must be revised in order to be checked and corrected. They also need to meet the rules of the journal. Please refer to the mdpi instructions for authors and also to the template. Some examples: Several names of the authors does not correspond to the number of the reference; References must appear by increasing numerical order along the text; use “XXXX and YYYY” when two authors exist and “XXXX et al.” when more than two authors exist; etc.

Specific Comment 5

Section 2

For the sake of the readers, all symbols must be properly defined in the text. In addition, well established equations must be justified, as much as possible, with references.

Specific Comment 6

Section 2.3

This section should be reorganized for better understanding of the readers. Also, equations are not numbered.

At line 256, please correct the double integral to a single one (“dx” instead of “dx dx”)…. The deflection is computed by integration of the curvature only once and along the “x” coordinate.

Specific Comment 7

Section 3.2 and Figures 2+3.

Please check if 180 mm is the width and 100 mm is the height of the test beams. It seems to me that there is a contradiction in the figures.

Specific Comment 8

Section 3.3

Please explain in the text the nomenclature used for the test beams.

Specific Comment 9

Section 4.1

It is stated that “F is the concentrated load of 3 kN”. However, in Figure 2, it is clear that the total load is splitted into two concentrated loads of 1.5 kN each located at thirds of the span. Please check this, including if the Equation (7) is valid for the actual loading setup.

Also, I don’t understand the presented values for the considered self-weight uniform load (25 kN/m) and for the cross-section moment of inertia (1.5E-5 m^4). For instance, for the self-weight, the calculation should be 0.1 m x 0.18 m x 25 (kN/m^3) = 0.45 kN/m, which is very different from the presented 25 kN/m…… Please check these values!

Specific Comment 10

Section 4.1

At line 331 and from Figure 7, I think it should be “two-thirds” instead of “one-third”. Please check!

Also, at line 335 and from Figure 7, I think it should be “less volatile” and not “more volatile”. Please check!

Specific Comment 11

Figure 10

Please. explain “&” in the legend.

Comments on the Quality of English Language

Please, see 1st specific comment for the authors.

Author Response

(The authors gave the same response as above.)

Reviewer 3 Report

Comments and Suggestions for Authors

The paper presents an interesting and useful study on the creep of HPC under temperature and humidity variations.

In general, the paper is well written, following a logical line. The introductory part is well referenced, presenting the necessity of this study and its novelty. Theoretical background is very well described and explained.

Experimental part needs to be revised / improved. The results obtained in practical tests are inline with the present literature; they are well presented and explained, charts being self-explanatory. The bridge application is a good addition to verify the proposed model.

There are some comments and recommendations:

- move reference numbers before the end of a sentence (see line 30, 33, etc);

- abbreviations should be described at their first occurrence (see line 54 - C-S-H);

- formulas in lines 243, 251 - 257 should be numbered; if so, all the equations should be re-numbered;

- rename chapter 3 as "Materials and methods";

- lines 279 - 281 should be deleted;

- section 3.1. Materials should be revised: add more details about the test samples (proposed beams) at the beginning of the section; how many samples for each specimen were produced? why there are 2 beams tested in standard room and only 1 in natural environment?

- according to figure 2, the test setup is a 4-point bending model, not 3-point one; the beam's height is 100mm or 180mm? if 100mm is correct, please explain why did you choose this?

- figure 5(b) seems to present a laboratory environment... beams Bvv and Bcv were tested in a natural environment or in a lab simulating the natural conditions? it is not very clear... please revise;

- according to the experimental beams dimensions, values for q and I are wrong (lines 314 - 316)... please check;

- use the same notation in text and figures (Bcc not BCC, Bcv not BCV, etc) - see figure 6, 7, 10, etc;

- use "Days" for X-axis in figures 7, 10, 11, 12;

- conclusions should present the limitations of the study and their implications (if not already addressed in the discussion section); also, they offer suggestions on how the research can be expanded or improved;

- many Chinese references... a widely spread can improved the paper.

Comments on the Quality of English Language

There are a few grammar / spelling / formatig errors:

- missing spaces (especially between references and reference's number) - see lines 35 (Sakata[10], 48, 48,50, 51, all the simalar ones on the entire paper, line 285, etc);

- missing spaces between values and measure units: line 283, 294, table 1, figure 13(b), etc; figure 8 and 9 - Y-axis;

- wrong use of capital letter - see line 329 (Beam);

Author Response

(The authors gave the same response as above.)

Reviewer 4 Report

Comments and Suggestions for Authors

Dear Authors,
thank you for your paper focused on the influence of humidity and temperature on creep deformation in beams. The experimental program and measurements are presented. The achieved results were applied to a real bridge structure. The experimental results were done in the laboratory and in situ. The creak was measured on four beams.
My comments are:
- line 22, here is written "... four cities." - do you mean four "places", "part" or do you really mean four cities as four towns?
- line 66, here is "Neville [32] - all authors should be listed correctly, in this case "Nasser and Neville [32]",
- line 68, here is "Wallo [35] - is it "Wallo et al [35]"?
- line 70, here is "Jagerman [28]" - literature [28] is "Bazat, Kim, Panula" - what applies? It needs to be fixed.
- the literature must be cited correctly in the text, if there are several names, the first name must be given and then "et al.",
- lines 91-92 - literature [44] is Wang et al., Schwesinger is literature [45],
- formulas on page 6 - I recommend supplementing the numbering of the formulas,
- tab. 1 - the units of single "materials" are missing, is it in [kg]? [m]? [kg/m3]?
- use "where" after equations, not with a capital "W",
- line 584 - (Literature) you must follow the citation rules and state surname or full name, not name with surname abbreviation.-
The paper is well written, it is understandable, it presents the results of the authors of the article. I recommend a minor revision before publishing.
Best regards.

Author Response

(The authors gave the same response as above.)

Reviewer 5 Report

Comments and Suggestions for Authors

The manuscript is well written. I have the following minor suggestions:

1.       What do authors mean by ‘components’? Please specify.

2.       Line 60: “… and faster flow of C-S-H colloids.” Faster flow from where to where? Do you mean within the matrix? Please specify.

3.       Remove lines from 279-281.

4.       Please clearly specify the difference between BCC1, BCC2, BVV and BCV.

5.       Line 381-321: Please explain more.

6.       Figure caption of Figure#7: Please specify these three conditions clearly in the caption. 

Comments on the Quality of English Language

Minor editing of the English language required

Author Response

(The authors gave the same response as above.)

Round 2

Reviewer 2 Report

Comments and Suggestions for Authors

I received the revised version of the article with revised title “Investigating the Influence of Fluctuating Humidity and Temperature on Creep Deformation in High-Performance Concrete Beams: A Comparative Study between Natural and Laboratorial Environmental Tests”. The authors have improved the article according to my previous comments. Hence, I consider that the article can be accepted for publication in the present form.

Reviewer 3 Report

Comments and Suggestions for Authors

The paper is greatly improved according to the comments and suggestions of the reviewers. Now everything is much clear and easy to understand.

I agree with the authors' answers and understand their point of view regarding the number of beams, dimensions and testing setup.

Congratulations!

Comments on the Quality of English Language

- table 3: use "number of beams" not "beam number"